# Metabolic Syndrome—An Emerging Constellation of Risk Factors: Electrochemical Detection Strategies

**DOI:** 10.3390/s20010103

**Published:** 2019-12-23

**Authors:** Madhurantakam Sasya, K. S. Shalini Devi, Jayanth K. Babu, John Bosco Balaguru Rayappan, Uma Maheswari Krishnan

**Affiliations:** 1Department of Molecular Physiology, School of Medicine, Niigata University, Niigata-9518510, Japan; sasyakaranam@gmail.com; 2Centre for Nanotechnology & Advanced Biomaterials (CeNTAB), SASTRA Deemed-to-be University, Thanjavur 613401, India; shalu2403@gmail.com (K.S.S.D.); jayanthkaranam@gmail.com (J.K.B.); rjbosco@ece.sastra.edu (J.B.B.R.); 3School of Chemical & Biotechnology, SASTRA Deemed-to-be University, Thanjavur 613401, India; 4School of Electrical & Electronics Engineering, SASTRA Deemed-to-be University, Thanjavur 613401, India; 5School of Arts, Science & Humanities, SASTRA Deemed-to-be University, Thanjavur 613401, India

**Keywords:** metabolic syndrome, biomarkers, biosensor, electrochemistry, nanomaterials

## Abstract

Metabolic syndrome is a condition that results from dysfunction of different metabolic pathways leading to increased risk of disorders such as hyperglycemia, atherosclerosis, cardiovascular diseases, cancer, neurodegenerative disorders etc. As this condition cannot be diagnosed based on a single marker, multiple markers need to be detected and quantified to assess the risk facing an individual of metabolic syndrome. In this context, chemical- and bio-sensors capable of detecting multiple analytes may provide an appropriate diagnostic strategy. Research in this field has resulted in the evolution of sensors from the first generation to a fourth generation of ‘smart’ sensors. A shift in the sensing paradigm involving the sensing element and transduction strategy has also resulted in remarkable advancements in biomedical diagnostics particularly in terms of higher sensitivity and selectivity towards analyte molecule and rapid response time. This review encapsulates the significant advancements reported so far in the field of sensors developed for biomarkers of metabolic syndrome.

## 1. Introduction

Advances in the health sector have greatly improved the health and life span of people. However, modern technological advancements have also resulted in dramatic changes in the way of life of individuals from both the calorie intake and energy consumption perspectives. A gross imbalance between the calories consumed and energy spent has resulted in the emergence of a new set of metabolic and age-related disorders that makes an individual more susceptible to chronic diseases, which if not detected in the early stages can turn fatal [1]. Metabolic syndrome (MetS) is a condition that refers to a cluster of risk factors that arises due to several dysfunctional biochemical pathways, thereby predisposing an individual to various non-communicable diseases [2]. MetS has emerged as a major health concern worldwide in the recent decades and is possibly connected with the life-style changes in the modern era [3]. Several risk factors have been identified to be part of the constellation of abnormalities leading to MetS (Figure 1). These include elevated blood glucose, triglycerides, cholesterol levels, obesity, oxidative stress and blood pressure [4,5,6,7,8].

Although each of the abnormalities cited reduces the quality of life when present independently, in combination they lead to severe health problems with increased risk of mortality. An individual is diagnosed with metabolic syndrome when at least three or more abnormalities that interfere with the body’s normal functioning are present [8].

## 2. Diagnosis of Metabolic Syndrome (MetS)

MetS is a complex, multi-factorial condition that pre-disposes an individual to several severe complications like cancer, cardiovascular diseases, chronic kidney diseases and neurodegenerative disorders [9]. Diagnosis of MetS is complicated as there are many risk factors that are associated with MetS. The identification of new inter-relationships between the factors has led to inclusion of new risk parameters for MetS. However, contradictory results from some studies have led to the elimination of several factors from the risk list. As a result, the definition of MetS has constantly evolved over the years. A scan of literature reveals that the first description of inter-relations between diabetes and hypertension was made during 1915–1916 by the physicians Hitzenberger and Rittner-Quittner [5,10]. Later, Kylin described the common mechanisms involved in the development of hyperglycemia, hypertension and hyperuricemia suggesting that these conditions arise due to common risk factors. During the late 1940s and early 1950s, several researchers identified obesity as the chief cause leading to the development of diabetes, hypertension, atherosclerosis, gout and dyslipidemia [5]. The nomenclature for this cluster of diseases caused due to inter-dependent dysfunctional metabolic pathways has evolved over the years and by common consensus is referred to as metabolic syndrome in the modern era (Table 1).

A large number of studies carried out on populations of different ethnicities, races, genders, ages and life-style habits have led to the evolution of the criteria to define MetS. MetS is a constellation of disorders and thus far only independent conditions have been detected using sensors. Individual quantification of glucose, triglycerides and superoxide has proved inadequate for accurate diagnosis of MetS. Combination of conditions that predispose an individual to MetS and the criteria for diagnosis of MetS has constantly been modified since the later 1990s. These are tabulated in Table 2. Attempts to develop multi-analyte panels using electrochemical methods are under active research currently and it may soon represent the next generation of smart diagnostic strategy for MetS in the future.

The guidelines for diagnosis of MetS as prescribed by IDF has been challenged by several research groups with respect to the cut-off values for the parameters as well as with respect to the parameters itself. The American Diabetes Association (ADA) in 2009 had challenged the inclusion of waist circumference as a criteria of MetS because large variations have been reported between individuals across different ethnicities and body stature [6]. Currently, the revised guidelines proposed by National Cholesterol Education Program—Adult Treatment Panel-III NCEP-ATP-III is being followed world-wide for diagnosis of MetS. But there are several caveats in this method also. Definition of obesity with respect to waist circumference measurements has not been universally accepted as there are many deviations reported [11]. Exclusion of several other quantifiable factors has been questioned by many groups. In 2010, the World Health Organization (WHO) had claimed that MetS is more of an educational concept rather than of clinical value. But, the recent consensus that has emerged is that MetS is a pre-disease condition that has emerged as a new age malady which makes an individual prone to many non-communicable diseases and reduces the survival rate [12]. The search for the most suitable predictors of MetS is still on earnestly across the globe.

The treatment modalities for MetS need to be tailored to cure the abnormalities presented by the individual. In other words, if two individuals are diagnosed with diabetes, the treatment regimen will differ based on the existence of other abnormalities in each individual. This necessitates accurate diagnostic tools for measurement of the risk factors. The following section details the advancements in the diagnostic field for major parameters of MetS.

## 3. Sensors for Metabolic Syndrome

As MetS is characterized by many risk factors that may vary from one individual to another, conventional methods have employed qualitative and quantitative methods to detect each risk factor independently. Distinct biomarkers found in body fluids or tissues that indicate an abnormal condition have to be chosen to identify each risk factor specifically. Conventional methods of diagnosing risk factors for metabolic syndrome involve measurement of individual parameters such as glucose, triglycerides, HDL and blood pressure (hypertension) apart from measurement of the waist circumference and body mass index (BMI) of the individual [13]. If at least three parameters are found to be abnormal, the person is diagnosed with MetS. But this entire process takes a long time; moreover, quick analysis is not possible using conventional strategies.

Most of the advances in detection of MetS complications have happened in the diagnosis of diabetes. Conventional methods for diagnosis of diabetes involve measurement of glycated hemoglobin levels and glucose levels at various time intervals during the day employing enzymatic and colorimetric assays. Hand-held glucometers based on enzymatic detection of glucose are also available commercially [14]. However, the enzyme-based strips cannot be reused and hence the method becomes expensive. Efforts to identify new methods to expand the detection range, reduce the response time and reuse the sensing element are underway in different parts of the globe. The strategies for the diagnosis of cardiovascular diseases include measurement of lipid profile comprising cholesterol, high-density lipoprotein (HDL), low-density lipoprotein (LDL), very low-density lipoprotein (VLDL), and triglycerides to identify dyslipidemia conditions. These analyses are carried out using biochemical assays that require higher sample volume and longer analysis time. In addition, plaque deposits arising due to atherosclerosis or vascular abnormalities are diagnosed employing imaging techniques such as X-rays, echocardiograms, electro-cardiograms (ECG), computed tomography (CT) scans, magnetic resonance imaging (MRI), laser dopplers and angiography. All these methods are highly expensive, time-consuming, require large and sophisticated instrumentation that are not portable and also need trained personnel for recording and analysis [15].

Renal failure has been conventionally diagnosed using biochemical assays for analyzing blood serum for creatinine levels and urea nitrogen [16]. Urine output and a total urine analysis (urinalysis) comprising a combination of biochemical assays to quantify urine protein and albumin, microscopic and visual examination to qualitatively detect abnormalities in colour and transparency as well as microbiological examination is carried out to detect renal dysfunction. Glomerular filtration rate and creatinine clearance rates are measured using radioisotopes or cystatin C [17]. Imaging techniques such as ultrasound and CT scan are also employed in conjunction with other examinations for diagnosis of renal disorders [18]. Waist circumference measurements, height and body weight measurements are carried out to determine body mass index and classify obesity [19]. In addition, heart rate, blood pressure monitoring and biochemical assays to determine cortisol and thyroid function are also prescribed by clinicians to diagnose obesity [20].

Efforts to develop strategies for accurate measure of distinct biomarkers in each condition will enable more accurate diagnosis of MetS-associated complications. Concerted efforts to identify quantifiable parameters that truly reflect the magnitude and severity of MetS complications are underway. Sensors with biological recognition element for the detection analytes have been successfully employed for the analysis of biomarkers in recent years [21]. Such sensors are called ‘biosensors’ and they commonly employ antibodies, enzymes, oligonucleotides, aptamers and cells as the recognition element. The first biosensor that appeared in the market in 1974 was a glucose analyzer originally developed by Clarke and Lyons in 1962 which was based on the formation of gluconic acid and hydrogen peroxide from glucose by the recognition element which was an enzyme glucose oxidase [22]. The sensor was then described as ‘enzyme electrode’. Since then, rapid advances have been made in the field of biosensors and it has now become an indispensable aspect of clinical diagnostics.

The transducer output from the sensor could arise due to changes in the optical properties that include absorbance, fluorescence, reflectance, refractive index, luminescence and surface plasmon resonance (SPR) or electrochemical properties where the signal is generated as a result of electron transfer due to redox reactions involving the analyte [23]. Calorimetric sensors measure heat changes occurring due to the reactions involving the analyte while electrical sensors respond to changes in electrical resistance. Piezoelectric sensors employ pressure-sensitive elements that record minute changes in the mass on the sensing element and generate a corresponding electrical signal [23]. Newer transduction strategies based on magnetic transduction principles are also on the anvil [24].

Among the diverse sensor categories, optical and electrochemical transduction mechanisms have been widely explored for quantitative detection of clinically relevant markers. Electrochemical biosensors are superior than optical sensors as they produce a stable signal towards the analyte. Furthermore, the electrochemical sensor modification process is easy and also cost-effective [25]. Hence, this type of electrochemical biosensor forms the focus of our review. A literature search was performed using the keywords “multi-analyte detection, electrochemical sensors, biomarkers for metabolic syndrome, MetS” in the databases like Scopus, PubMed, Google Scholar and SciFinder. The relevant articles were chosen for the review.

## 4. Evolution of Electrochemical Biosensors

Electrochemical biosensors have been developed over the years in terms of their superior sensitivity, quick response time as well as the wide range of analyte molecules that can be detected easily using the surface modification process according to the analyte of interest. Figure 2 depicts the schematic representation of the different generations of electrochemical sensors developed over the years. The earliest electrochemical sensors known as the first-generation sensors quantified the analyte indirectly by measuring the accompanying changes in the hydrogen peroxide or oxygen concentrations. However, this strategy restricted the type of analytes that could be detected as well as that had lower sensitivity. As the redox site in most enzymes is buried deep in the interior, accessibility restrictions and slow electron transfer from the analyte are common in these types of sensors resulting in slow response time [8,14]. These limitations were overcome in the second-generation sensors that employed a redox mediator such as ferrocene or quinone to shuttle electrons from the bio-recognition element to the electrode surface. This strategy eliminates indirect measurement of the analyte and enables faster detection. However, use of a redox mediator for electron transfer makes these types of sensors prone to interference from other electro-active species in the sample. Furthermore, choice of an appropriate redox mediator is an added complexity in the sensing process [26]. Hence, the third-generation sensors employed a direct transfer of the electrons from the biorecognition element to the working electrode surface. To ensure rapid response and sensitivity in this type of sensors, intimate contact of the biorecognition site with the electrode surface is required. The immobilization of the biosensing element on the working electrode surface, therefore, acquires prime importance for efficient electron transfer. Both physical and chemical methods of immobilization have been reported. The interactions between the immobilized biomolecule and the electrode surface are generally hydrogen bonding, electrostatic, hydrophobic and dipole–dipole interactions. However, the third-generation sensors are limited by accessibility issues, diffusional limitations and denaturation risks of the biological sensing element [27]. Hence recent times have seen the emergence of a fourth generation of sensors which employ chemical entities that serve as bio mimics to overcome the stability issues associated with the immobilization of biological molecules on the electrode surface [28].

Multiple enzymes can be immobilized into each substrate present in the array which would enable the simultaneous detection of MetS biomarkers in a label-free approach. The advantage of these sensors will be quick response time and low sample volume compared to conventional sensors focusing on single analyte detection. However, matching the selectivity of biological molecules remains a major challenge in the bio-mimetic sensing elements. Instead of conventional electrodes researchers have attempted several strategies using screen-printed electrodes for the analysis of metabolites [29,30,31]. In recent years, response time of the electrochemical sensors has been improved by the incorporation of nanomaterials as interface between the electrode and enzyme surface. These nano materials further enhance the surface coverage as well as sensitivity resulting in low detection limits and low sample requirements [32]. A wide range of nanomaterials such as metal nanoparticles [33,34,35], metal oxide nanostructures [36,37,38,39,40,41,42,43,44,45,46], polymeric nanoparticles [47], carbon nanotubes [48,49,50,51], graphene [52,53,54,55,56], quantum dots [57,58,59,60,61], hydrogels [62,63,64,65,66], and ceramic nanostructures [67,68] have been extensively studied as interface materials in electrochemical nanosensors. The nano-dimensional interface in electrochemical sensors exhibits several unique characteristics. Their high surface area-to-volume ratio facilitates immobilization of larger number of biomolecules thereby improving the number of active sites on the electrode surface [69]. Metal and metal oxide nanoparticles were proved to enhance the electron transfer rate due of their tiny size and higher conductivity. Furthermore, the close contact between the nanoparticle and biomolecule improves the stability of the biosensor [32]. In several instances, metallic nanoparticles have been reported to serve as catalysts for the electrochemical reaction involving the analyte [70]. The introduction of more than one type of nanoparticle as interface material will further leads to the enhancement of biosensor properties in terms of wide linear range, quick response time and high sensitivity [71]. Such smart sensor systems with hybrid nano interfaces for multiple analyte detection will ensure for rapid and accurate analysis of biomarkers in the field of clinical diagnosis in the modern era.

## 5. Electrochemical Detection Strategies for Multi-Analyte Detection

Development of sensors for single analytes may not always be conclusive for diagnosis of a clinical anomaly, especially for MetS where co-existence of multiple complications necessitates the development of integrated systems that can simultaneously detect multiple analytes. Although there are plenty of reports available for single analyte detection, only a miniscule fraction of the literature is available for multi-analyte detection using electrochemical methods. As MetS is a constellation of disorders, a single biomarker is insufficient to diagnose it. Therefore, multi-analyte detection is required for diagnosis of MetS. Conventional diagnosis also uses multiple parameters to diagnose MetS. However, these parameters are subjective and less sensitive, thereby requiring multiple tests over an extended duration for diagnosis. The sensor arrays based on electrochemical detection and nano-interfaces can detect multiple biomarkers rapidly with high sensitivity. Fabrication of these sensors on disposable and cheap substrates like paper can enable affordable low-volume diagnosis and large-scale screening of populations for multiple dysfunctions associated with MetS such as elevated blood glucose levels, triglycerides, cholesterol levels, obesity, oxidative stress and hypertension. Electrochemical biosensors with highly specific detection capabilities eliminate interferences from other metabolites encountered in body fluids. A positive result for more than one marker in the panel indicates higher risk of MetS and the individual can then be referred for additional tests to confirm the preliminary diagnosis. Such early detection strategies can reduce mortality and improve quality of life. But, the current challenge is to identify the most relevant biomarkers that truly represent the risk of MetS and integration of these marker panels in a single array without compromising on the specificity as well as detection range. Another pitfall in such strategies is that they are invasive methods of diagnosis though the sample volume required may be low. However, the benefits of such interventions outweigh their limitations and hence such strategies can be invaluable in the near future for clinical applications.

Employment of hybrid nano-interface materials for multiple-analyte detection has been proved to improve sensor performance as well as reduce the response time [72,73]. The techniques like CV (cyclic voltammetry), Amp (amperometry), CA (Chronoamperometry), EIS (Electrochemical impedance spectroscopy) and DPV (differential pulse voltammetry) are among electrochemical methods that provide different information on the interactions between the analyte and sensing element at the electrode-electrolyte interface. CV is extensively employed technique in electrochemical sensors which gives insights in to the reaction that occurs at the electrode whereas DPV is a pulse technique with better sensitivity than CV. It displays a single pulse of either an oxidation or a reduction process occurring in the system on the introduction of the sample containing the analyte of interest. The amperometric technique is another sensitive method that monitors the current changes with time at a constant potential. The step-wise profile obtained in dynamic conditions provides information on the stability and the reproducibility of the system. Electrochemical impedance provides information on the capacitance changes at the electrode surface accompanying changes in the concentration of the analyte of interest. Choice of a technique therefore, depends on the combination of electrode, electrolyte, nanointerface and capture agent employed. The major contributor to improved sensitivity is from the nanointerface materials and the enzymes/antibodies used for specific detection while the contribution by advanced instrumentation and techniques are minimal.

Dopamine, a neurotransmitter, has been observed to be involved in the regulation of glucose and lipid metabolism, blood pressure and insulin release [74]. Obese individuals have been found to exhibit reduced levels of dopamine [75]. Obese individuals and diabetes-complicated individuals are also reported to have depleted levels of ascorbic acid [76,77]. Uric acid is a well-established marker for kidney disorders [78,79]. Hence, these analytes can be useful for prediction of MetS risk in individuals. A glassy carbon electrode (GCE) modified with ZnO was employed for the simultaneous detection of dopamine, ascorbic acid and uric acid [80] in the detection range of 6–960 µM,15–240 µM and 0.5–800 µM for dopamine (DA), ascorbic acid(AA) and uric acid (UA), respectively. Similarly, as shown in Figure 3D a combination of carbon black–carbon nanotube nano-interface and polyimide has been employed as a working electrode for the simultaneous sensing of AA, DA, and UA. This fabricated sensor showed the enhanced sensitivity to only two of the analyte’s DA and UA, with lowest detection limit of 1.9 µM and 3 µM respectively. Applicability of the sensor was tested in human urine samples with good recovery values [81]. Another sensor with a simplified interface comprising carbon black-chitosan mixture was tuned as a water soluble homogenous ink deposited on a GCE for concurrent detection of AA, UA and DA as shown in Figure 3A with a lower detection limit of 0.1 μM achieved for all the three analytes. Though not developed specifically for MetS, this sensor was tested in real samples such as vitamin C tablets for AA, dopamine chloride injection for DA, and human urine samples for UA, and has potential to be used as a MetS screening device [82].

Apart from glucose, glutamine and glutamate have been identified as key metabolites in glucose metabolism while lactate accumulation has been cited as a key indicator of metabolic disorders. A disposable electrochemical biosensor precalibrated in 1–5µL flow-through cell was fabricated for the simultaneous detection of glucose, lactate, glutamine and glutamate [83]. This sensor exhibited a detection range upto 5 mM for glucose, 2 mM for lactate, 1 mM for glutamine and 200 µM for glutamate. An electrochemical array chip based sensor has been developed for the simultaneous detection of glucose and lactate [79]. The working electrode of this sensor comprises iridium oxide. The electrochemical detection of glucose was linear between the concentrations of 5 mM and 10 mM while the linear range for lactate measurement was found to be 0–2 mM. Sol-gel based mesoporous silica combined with multiwalled carbon nanotubes (MWCNTs) was employed as a hybrid interface that was successfully employed for the detection of dopamine, uric acid and paracetamol in biological samples [84]. Though paracetamol is a drug that has no direct correlation with MetS, the dopamine and uric acid levels in the body provide important information about MetS-associated disorders. The working electrode used in this sensor was modified with a hybrid nano interface of SiO_2_ and multi walled carbon nanotubes. The detection range for uric acid, dopamine and paracetamol was found to be 0.6–4.65 µM, 0.13–4.64 µM and 0.67–4.65 µM, respectively. A bienzymatic sensor for the simultaneous detection of glucose (Glu) and cholesterol (ChL) has been designed using polythionine and gold nanoparticles incorporated between the enzymes glucose oxidase and cholesterol oxidase on a glassy carbon electrode [85]. The amperometric measurement of glucose and cholesterol in the samples was achieved in the concentrations of 0.008–6 mM and 0.002–1 mM respectively. It has been shown that the use of iron oxide nano particles enhances the binding efficiency of the enzyme cholesterol oxidase for the detection of cholesterol and glucose oxidase for glucose detection [86]. Along similar lines, a low-cost biosensor for monitoring the glucose (Glu), UA, and cholesterol (ChL) simultaneously was fabricated using a flexible microneedle electrode consisting of gold/titanium film on the surface along with electrodeposited polyaniline nanofibers/platinum nanoparticles for electron transfer and respective enzymes on the modified film. Samples were prepared by spiking known amounts of Glu, UA and ChL in fetal bovine serum albumin and detected using amperometric technique. The reported microneedle sensor enables self-health monitoring of blood metabolites by using minimally invasive microneedles that permeate up to the dermal layer of the skin [87]. A multi-analyte sensor for the detection of seven analytes namely, H_2_O_2_, lactate, NADH (reduced nicotinamide adenine dinucleotide), ascorbic acid, uric acid, nitrite and dopamine was fabricated with a hybrid interface of iron oxide nanoparticles and reduced graphene oxide nanosheets on an indium tin oxide electrode [88]. While the role of H_2_O_2_, lactate, ascorbic acid, uric acid and dopamine in MetS-related disorders have already been identified, NADH being a key co-enzyme in glucose and lipid metabolic pathways are important indicators of the metabolic regulation in an individual [89]. Similarly, it has been demonstrated that reduced nitrite and nitrate levels contribute to endothelial dysfunction and MetS [90]. The amperometric measurement of each analyte was accomplished at specific potentials of −0.3 V (H_2_O_2_), +0.01 V (ascorbic acid), +0.1 V (lactate), +0.05 V (NADH), +0.33 V (uric acid), +0.16 V (dopamine) and +0.7 V (nitrite). The sensor was able to quantify lactate levels between 0. 2 µM and 2.2 mM, uric acid levels in the range4–20 µM, ascorbic acid in the sample between 160 µM and 7.2 mM, dopamine concentrations of0.4–3.5 µM, H_2_O_2_ concentrations between 20 nM and 0.28 µM, nitrite and NADH levels in the range 20–210 µM and 2–15 µM respectively [88]. Magnetite nanoparticles coated with polydopamine were covalently attached to glucose oxidase as shown in Figure 3E was employed for the analysis of glucose in human serum samples [89].In another interesting work, molybdenum disulphide (MoS2) nanosheets supported Au–Pd bimetallic nanoparticles were employed for the enzyme-free sensing of glucose and H_2_O_2_ as indicated in Figure 3B. This non-enzymatic sensor exhibited a wide linear range of 0.8 μM–10 mM for H_2_O_2_ and 0.5–20 mM for glucose [91]. As shown in Figure 3C glucose and uric acid were simultaneously monitored using screen printed strip-based electrode with dual channels. One channel was modified with glucose oxidase and the second channel is modified with uric acid oxidase for specific detection of glucose and uric acid simultaneously from a single sample [92]. Several attempts to develop sensors for the quantification of glucose and hydrogen peroxide are reported in literature. Palladium-cobalt nanoparticles over carbon nanotubes using one-pot synthesis method was reported by Huang et al. as shown in Figure 3F [93].

An electrochemical biosensor comprising nanoporous nanosponge architecture of Pd–Cu alloy fabricated by etching off Al from PdCuAl alloy was employed for the detection of glucose and H_2_O_2_. This enzyme-free sensor exhibited long-term stability, high sensitivity and wide linear range of 1–30 mM for glucose and 0.1–2 mM for H_2_O_2_ [94]. Electrospun Co3O4 nanofibers were used for electrochemical analysis of glucose and H_2_O_2_ by cyclic voltammetry, chronoamperometry and impedance analysis in an alkaline medium of 0.1 M NaOH as supporting electrolyte. The sensor exhibited a short response time ~1.5 s for glucose and 6.6 s for H_2_O_2_ and a linear range of 50–1000 µM and 20–400 µM for glucose and H_2_O_2_, respectively [95]. A novel electrodeposition method was employed for the synthesis of snowflake-like Pt–Pd bimetallic nano-clusters on screen-printed gold nano film electrode (SPGFE). This enzyme less sensor was employed for the simultaneous detection and quantification of glucose and H_2_O_2_ [30]. The sensor performed well in the concentrations 0–16 mM for glucose and 0.005–6 mM for H_2_O_2_. An amperometric biosensor array comprising of phenylenediamine modified transducers was fabricated for the detection of choline, glutamate, glucose, lactate, acetylcholine and pyruvate in CSF (cerebrospinal fluid) and blood plasma within a minute and exhibiting a linear range prevailing between 0.001–0.01 and 0.2–2.5mM and limit of detection (LOD) of 1–5 µM [96]. Several other combinations of nanointerfaces and electrode materials have been reported for the detection of glucose and H_2_O_2_ that are summarized in Table 3. All these sensors are still in the research phase and are yet to be deployed in a clinical set-up. It also lists the details of electrochemical sensors reported for sensing of multiple analytes since 2008 for biomarkers associated with MetS complications. Although each component in some combinations such as acetaminophen and codeine are not directly associated with MetS, the other analytes detected in the combination provide insights in to the severity of MetS disorders in the individual.

## 6. Conclusions and Future Directions

Metabolic syndrome is rapidly emerging as a major health risk in the world. Effective and rapid diagnosis of this condition requires multi-analyte detection. Electrochemical sensors have emerged as a major tool for rapid monitoring of disease markers. Although these sensors have not been specifically designed for diagnosis of metabolic syndrome, their efficiency in detecting the biomarkers that are associated with some of the complications currently identified as part of metabolic syndrome suggests that these sensors could usher in superior devices for diagnosis of metabolic syndrome. Although the importance of multi-analyte detection has been recognized, not many effective sensor panels have been developed for diagnosis of MetS complications. The scan of literature reveals several interesting attempts to fabricate multi-analyte electrochemical sensors employing different types of nanointerface and biological-sensing elements. Therefore, development of sensor arrays employing an intelligent combination of nanointerfaces, enzymes and electrode fabrication for specific detection of key markers from biological fluids that can predict the risk of MetS is of current importance. Currently, this field is in its infancy and further innovations in development of sensor arrays for quantification of specific markers associated with MetS is on the anvil in coming decades. Recent advances in fabrication of low-cost and disposable electrodes using 3D printing and screen-printing methods, and use of machine learning and artificial intelligence to predict MetS associated complications could be a game changer in the future for large-scale screening of populations.

## Figures and Tables

**Figure 1 sensors-20-00103-f001:**
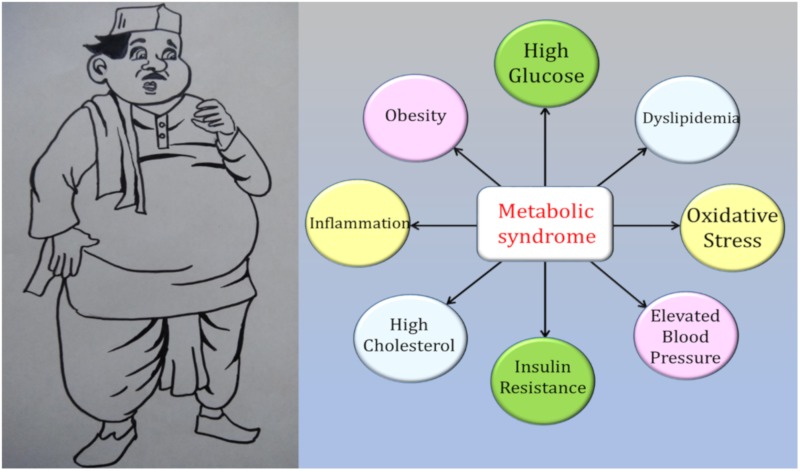
Schematic representation of the risk factors for metabolic syndrome.

**Figure 2 sensors-20-00103-f002:**
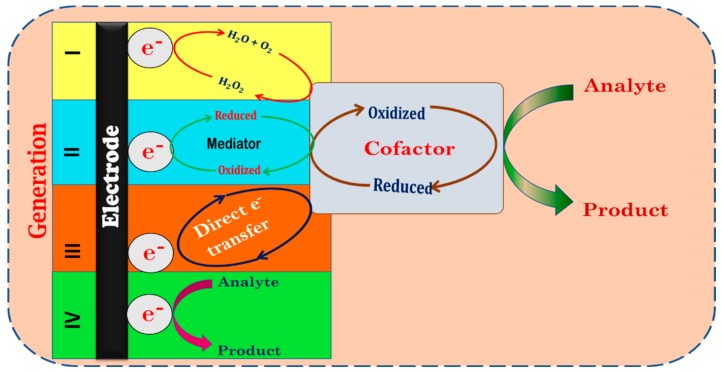
Schematic representation of various generations of electrochemical sensors.

**Figure 3 sensors-20-00103-f003:**
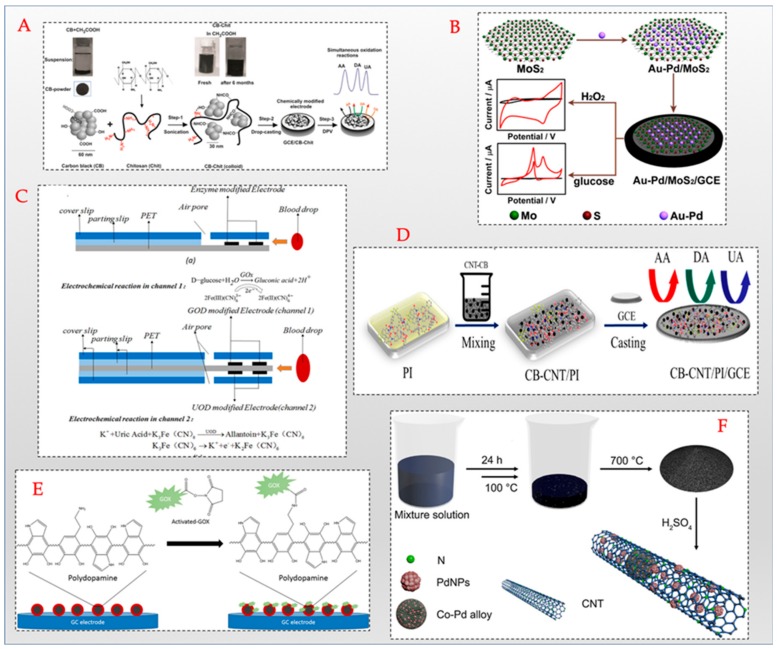
Schematic representation of different modification processes of working electrodes utilizing different nanomaterials (**A**) carbon black [82] Copyright © 2017 Elsevier, (**B**) molybdenum disulphide (MoS2) nanosheets [91] Copyright © 2017 Elsevier, (**C**) enzyme modified screen printed electrodes [92] Copyright © 2017 Elsevier, (**D**) carbon black–carbon nanotubes [82], (**E**) magnetite nanoparticles/polydopamine [89] Copyright © 2017 Elsevier, (**F**) Palladium-cobalt nanoparticles/ carbon nanotubes [93] Copyright © 2017 Elsevier.

**Table 1 sensors-20-00103-t001:** Different nomenclature given to metabolic syndrome (MetS) conditions over the years.

Year	Nomenclature	Risk Factors Included	Proposed By
1923	Hypertoni–Hyperglycemi–Hyperurikemi syndrome	Hypertension, hyperglycemia, hyperurecemia	Kylin
1966	Trisyndrome metabolique	Gout, diabetes, hyperlipidemia	Camus
1967	Plurimetabolic syndrome	Hyperlipidemia, obesity, diabetes, hypertension, coronary heart disease	Avogaro and Crepaldi
1968	Wohlstands-syndrom (Syndrome of affluence)	Hyperlipidemia, obesity, diabetes, hypertension, coronary heart disease	Mehnert and Kuhlmann
1981	Metabolische-syndrom (Metabolic syndrome)	Hyperlipidemia, hyperinsulinemia, obesity, diabetes, hypertension, gout, thrombophilia	Hanefeld and Leonhardt
1988	Syndrome X	Impaired glucose tolerance, hyperinsulinemia, very low-density lipoprotein (VLDL), triglycerides, cholesterol, hypertension, low high-density lipoprotein (HDL)	G.M. Reaven
1989	Deadly quartet	Central adiposity, impaired glucose tolerance, hypertriglyceridemia, hypertension	Kaplan
1991–1992	Insulin resistance syndrome	Insulin resistance, diabetes, hypertriglyceridemia	DeFronzo and Ferranini, Haffner
1994	Visceral fat syndrome	Visceral fat, diabetes, dyslipidemia	Nakamura and Matsuzawa

**Table 2 sensors-20-00103-t002:** Various diagnostic criteria used for MetS.

Agency	Risk Factor					
	Body Weight	Insulin Resistance	Lipids	Blood Pressure	Glucose	Others
World Health Organization (WHO), 1998	Waist/hip >0.9 (men) >0.85 (women) or body mass index (BMI) >30 kg/m^2^	IGT/IFG/type 2 diabetes or lower insulin sensitivity + any 2 of the other factors	TG ≥150 mg/dL and/or HDL <35 mg/dL (men) <39 (women)	≥140/90 mm Hg	IGT/IFG/type 2 diabetes	Micro-albuminuria Urinary excretion rate >20 mg/min or albumin/creatinine >30 mg/g
European Group for the study of Insulin Resistance (EGIR), 1999	Waist circumference ≥94 cm (men) ≥80 cm (men)	Plasma insulin >75th percentile	TG ≥ 150 mg/dL and/or HDL <39 mg/dL	≥140/90 mm Hg	IGT/fasting plasma glucose >110 mg/dL	None
National Cholesterol Education Programme/Adult Treatment Panel III (NCEP/ATP III), 2001	Waist circumference ≥102 cm (men) ≥8 cm (men)	Any three of the five factors listed	TG ≥150 mg/dL and/or HDL <40 mg/dL (men) <50 (women)	≥130/85 mm Hg	>110 mg/dL	None
American Association of Clinical Endocrinologists (AACE), 2003	BMI ≥25 kg/m^2^	IGT/IFG + any of the other factors	TG ≥ 150 mg/dL and/or HDL <35 mg/dL (men) <39 (women)	≥130/85 mm Hg	Fasting plasma glucose 110–126 mg/dL; post-prandial 140–200 mg/dL	None
International Diabetes Federation (IDF), 2005	Ethnicity based values for waist circumference >94 cm (Euro men) >80 cm (Euro women) >90 cm (Asian men) >80 cm (Asian women)	Not listed	TG ≥ 150 mg/dL and/or HDL <40 mg/dL (men) <50 (women)	≥130/85 mm Hg	>100 mg/dL	None

Impaired Glucose Tolerance (IGT); Impaired Fasting Glucose (IFG); Triglycerides (TG); High density lipoprotein (HDL).

**Table 3 sensors-20-00103-t003:** Electrochemical sensors reported for sensing of dual and triple analytes reported.

	Analyte	Nano-Interface	Enzymes Used	Technique	Ref.
Dual Analytes	Glucose and H_2_O_2_	Pt–Pd bimetallic clusters	Yes	CV, Amp	[30]
	Glucose and Cholesterol	Poly-thionine film	No	CV, Amp	[85]
	Glucose and H_2_O_2_	Au–Pd bimetallic nanoparticles	No	CV, Amp	[91]
	Glucose and Uric acid	Carbon ink	Yes	CA	[92]
	Glucose and H_2_O_2_	Pd-CoCNTs	No	CV, Amp, EIS	[93]
	Glucose and H_2_O_2_	PdCu alloy	No	CV, Amp	[94]
	Glucose and H_2_O_2_	Co_3_O_4_	No	CV, Amp	[95]
	Glucose and H_2_O_2_	Cu_2_O	No	CV, Amp, EIS	[97]
	Glucose and H_2_O_2_	Silver–DNA hybrid nanoparticles	Yes	CV, Amp	[98]
	Glucose and H_2_O_2_	CuO/rGO/Cu_2_O	No	CV, Amp	[99]
	Glucose and Maltose	MWCNTs	No	CV, Amp	[100]
	Glucose and Urea	E-DNA	No	CV, Amp, EIS	[101]
	Glucose and H_2_O_2_	Perovskite	No	Amp	[102]
	Glucose and H_2_O_2_	CoS	No	CV, Amp, EIS	[103]
	Glucose and H_2_O_2_	Graphene wrapped CuO nanocubes	No	CV, Amp	[104]
	Glucose and H_2_O_2_	Ag nanowires-CS	Yes	CV, Amp	[105]
Triple Analytes	Uric Acid, Dopamine, Ascorbic Acid	Carbon black–carbon nanotube/polyimide composite	No	CV, DPV, Amp	[81]
	Ascorbic Acid, Dopamine and Uric Acid	Water-soluble homogenous carbon black–chitosan ink	No	CV, DPV, Amp	[82]
	Glucose, Uric Acid, Cholesterol	Gold/titanium electrodeposited with polyaniline on platinum nanoparticles	Yes	Amp	[87]
	Ascorbic acid, Dopamine and Uric acid	Gold electrode patterned on polymethylmethacrylate	No	CV, DPV	[106]
	Glucose, Ethanol and Cholesterol	Polydopamine-coated magnetic nanoparticles	Yes	CV, Amp	[89]
	Glucose, D-Fructose, Sucrose	3-D Cu foam	No	CV, A	[107]

CV: cyclic voltammetry, A: amperometry, EIS: electrochemical impedance spectroscopy, DPV: differential pulse voltammetry, H_2_O_2_.hydrogen peroxide, Glu= glucose, Ag= silver, E-DNA= oligo nucleotide probe-based electrochemical DNA, Co_3_O_4_= cobalt oxide, Pt–Pd= platinum palladium bimetallic nanoparticles, MWCNT= multiwalled carbon nanotube, CuO=copper oxide, rGO= reduced graphene oxide, CoS= cobalt sulfide.

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
