# Peer review of "Metabolic Syndrome—An Emerging Constellation of Risk Factors: Electrochemical Detection Strategies"

_sensors, 2019, doi:10.3390/s20010103_

Round 1
Reviewer 1 Report
Its a deep and well structured review about sensors that potentially could be used as the screening for metabolic syndrome (MetS). However, MetS is a multifactorial and multi-symptom condition, so the clinical scenario where and how these sensors could be used was not discussed in the text.
My major concerns are:
1.- In the introduction include the methodology for the reference reviewed.
2.- Discuss how the multi-analytie sensors could be used in the clinical scenario for MetS screening (population, symptoms, conditions, test for substitution, etc) and their main benefits and limitations.
3.- Discuss the status of the sensors reviewed (in research, in a commercial device, etc).
* reference 3 was retracted, I suggest to cite the sources.
Author Response
Reviewer 1
Its a deep and well structured review about sensors that potentially could be used as the screening for metabolic syndrome (MetS). However, MetS is a multifactorial and multi-symptom condition, so the clinical scenario where and how these sensors could be used was not discussed in the text.
My major concerns are:
1.In the introduction include the methodology for the reference reviewed.
A literature search was performed using the keywords “multi-analyte detection, electrochemical sensors, biomarkers for metabolic syndrome, MetS” in the databases like Scopus, Pubmed, Google Scholar and Scifinder. The relevant articles were chosen for the review. This statement is now included in the revised manuscript for better clarity. The technical methods used in these sensors for quantification of the MetS marker have been tabulated in Table 2 in the manuscript.
Discuss how the multi-analytie sensors could be used in the clinical scenario for MetS screening (population, symptoms, conditions, test for substitution, etc) and their main benefits and limitations.
We thank the reviewer for the suggestion. The following paragraph has been included in the revised manuscript as suggested:
As MetS is a constellation of disorders, a single biomarker is insufficient to diagnose it. Therefore, multi-analyte detection is required for diagnosis of MetS. Conventional diagnosis also uses multiple parameters to diagnose MetS. However, these parameters are subjective and less sensitive thereby requiring multiple tests over an extended duration for diagnosis. The sensor arrays based on electrochemical detection and nano-interfaces can rapidly detect multiple biomarkers with high sensitivity. Fabrication of these sensors on disposable and cheap substrates like paper can enable affordable low volume diagnosis and large scale screening of populations for multiple dysfunctions associated with MetS such as elevated blood glucose levels, triglycerides, cholesterol levels, obesity, oxidative stress and hypertension. Electrochemical biosensors with highly specific detection capabilitieseliminate interferencesfrom other metabolites encountered in body fluids. A positive result for more than one marker in the panel indicates higher risk of MetS and the individual can then be referred for additional tests to confirm the preliminary diagnosis. Such early detection strategies can reduce mortalities and improve quality of life. However, the current challenge is to identify the most relevant biomarkers that truly represent the risk of MetS and integration of these marker panels in a single array without compromising on the specificity as well as detection range. Another pitfall in such strategies is that they are invasive methods of diagnosis though the sample volume required may be low. However, the benefits of such interventions outweigh their limitations and hence such strategies can be invaluable in the near future for clinical applications.
Discuss the status of the sensors reviewed (in research, in a commercial device, etc). reference 3 was retracted, I suggest to cite the sources.
All the sensors reported in Table 3 are still in the research phase and are yet to be deployed in a clinical set-up. This has been included in the revised manuscript. We thank the reviewer for pointing outthe error.The reference 3 has been replaced in the revised manuscript.

Reviewer 2 Report
This review summarizes resent advancements in sensors for metabolic syndrome detection. The review is comprehensive and well organized. There is a little issue. The Figure 3d is not clear. Besides, it is better that the authors provide a future study insight for the reader.
Author Response
Reviewer 2
This review summarizes recent advancements in sensors for metabolic syndrome detection. The review is comprehensive and well organized. There is a little issue.
The Figure 3d is not clear.
We thank the reviewer for the valuable suggestion. Higher clarity image has now been incorporated in the revised version of the manuscript.
Besides, it is better that the authors provide a future study insight for the reader.
The following paragraph has been included in the revised manuscript as suggested by the reviewer.
Though the importance of multi-analyte detection has been recognized, not many effective sensor panels have been developed for diagnosis of MetS complications. Therefore, development of sensor arrays employing an intelligent combination of nanointerfaces, enzymes and electrode fabrication for specific detection of key markers from biological fluids that can predict the risk of MetS is of current importance. Currently, this field is in its infancy and further innovations in development of sensor arrays for quantification of specific markers associated with MetS is on the anvil in the coming decades. Recent advances in fabrication of low cost and disposable electrodes using 3D printing and screen printing methods,and use of machine learning and artificial intelligence to predict MetS associated complications could be a game changer in the futurefor large scale screening of populations.

Reviewer 3 Report
This manuscript described the evolution of the sensors from the first generation to the fourth generation of ‘smart’ sensors for Metabolic Syndrome. Metabolic syndrome is a condition that results due to dysfunction of different metabolic pathways leading to the increased risk of disorders such as hyperglycemia, atherosclerosis, cardiovascular diseases, cancer, neurodegenerative disorders etc. Chemical and biosensors capable of detecting multiple analytes may provide an appropriate diagnostic strategy for Metabolic syndrome. This manuscript provides a comprehensive view for multiple Electrochemical biosensor, but some issues must be addressed.
The abstract described “……the first generation to the fourth generation of ‘smart’ sensors” , it seemed that there are no clear define about the different generations of these biosensors. please explain it and add some figures about different generation of biosensors. what’s the advantages of the last generation biosensors for the detection of metabolic syndrome? Could you provide the road of the evolution of the biosensors for this disease. There are 107 references in this manuscript, but the reference [111] was listed in Table 3. Please add the related reference in the manuscript. Please improve the quality of Figure 3. Only part of Figure 3D is shown in the manuscript, please revise it. Different techniques (such as CV, Amp, CA, EIS and DPV, etc) were chosen as the electrochemical strategies for Metabolic Syndrome? Could you compare the advantages and disadvantages of these techniques for different analytes? Different nano-materials are utilized for the detection of different analytes. What about the sensitivity of these methods? The improved performance of these methods is due to the used nanomaterials or the improved instruments or other factors? The style of the references need to be unified.Author Response
Reviewer 3
This manuscript described the evolution of the sensors from the first generation to the fourth generation of ‘smart’ sensors for Metabolic Syndrome. Metabolic syndrome is a condition that results due to dysfunction of different metabolic pathways leading to the increased risk of disorders such as hyperglycemia, atherosclerosis, cardiovascular diseases, cancer, neurodegenerative disorders etc. Chemical and biosensors capable of detecting multiple analytes may provide an appropriate diagnostic strategy for Metabolic syndrome. This manuscript provides a comprehensive view for multiple electrochemical biosensor, but some issues must be addressed.
The abstract described “……the first generation to the fourth generation of ‘smart’ sensors” , it seemed that there are no clear define about the different generations of these biosensors. please explain it and add some figures about different generation of biosensors.
The evolution of biosensors has occurred through three generations for single analyte detection.These are as follows:
The earliest electrochemical sensors known as the first-generation sensors quantified the analyte indirectly by measuring the accompanying changes in the hydrogen peroxide or oxygen concentrations. In the second-generation sensors that employed a redox mediator such as ferrocene or quinone to shuttle electrons from the bio-recognition element to the electrode surface. The third-generation sensors employed a direct transfer of the electrons from the biorecognition element to the working electrode surface. The emergence of fourth generation of sensors which employ chemical entities that serve as bio mimics to overcome the stability issues associated with the immobilization of biological molecules on the electrode surface
The above details along with Figure 2 depicting the principle of detection in the different generations of sensors is incorporated in the revised manuscript as suggested.
What are the advantages of the last generation biosensors for the detection of metabolic syndrome?
Multiple enzymes can be immobilized into each substrate present in the array which would enable the simultaneous detection of MetS biomarkers in a label-free approach. The advantage of these sensors will be quick response time and low sample volume compared to conventional sensors focussing on single analyte detection.
Could you provide the road of the evolution of the biosensors for this disease?
The following information has been included in the revised manuscript for clarity.
MetS is a constellation of disorders and thus far only independent conditions have been detected using sensors. Individual quantification of glucose, triglycerides and superoxide have proved inadequate for accurate diagnosis of MetS. Combination of conditions that predispose an individual to MetS and the criteria for diagnosis of MetS has constantly been modified since later 1990s. These are tabulated in Table 2. Attempts to develop multi-analyte panels using electrochemical methods is under active research currently and it may soon represent the next generation of smart diagnostic strategy for MetS in the future.
There are 107 references in this manuscript, but the reference [111] was listed in Table 3. Please add the related reference in the manuscript.
We thank the reviewer for pointing out the error that had inadvertently crept in. The corrections have been incorporated in the revised manuscript.
Please improve the quality of Figure 3. Only part of Figure 3D is shown in the manuscript, please revise it.
We thank the reviewer for the suggestion. The quality of Figure 3D has been now improved and incorporated in the revised manuscript.
Different techniques (such as CV, Amp, CA, EIS and DPV, etc) were chosen as the electrochemical strategies for Metabolic Syndrome? Could you compare the advantages and disadvantages of these techniques for different analytes? What about the sensitivity of these methods?
The following information has been included in the revised manuscript for clarity:
The techniques like CV (cyclic voltammetry), Amp (amperometry), CA (Chronoamperometry), EIS (Electrochemical impedance spectroscopy) and DPV (differential pulse voltammetry) are part of electrochemical methods that provide different information on the interactions between the analyte and sensing element at the electrode-electrolyte interface. CV is extensively employedtechnique in electrochemical sensors which gives insights in to the reaction that occurs at the electrode whereas DPV is a pulse technique with better sensitivity than CV. It displays a single pulse of either oxidation/reduction process occurring in the system on introduction of the sample containing the analyte of interest. Amperometric technique is another sensitive method that monitors the current changes with time at a constant potential. The step-wise profile obtained in dynamic conditions provide information on the stability and the reproducibility of the system. Electrochemical impedance provides information on the capacitance changes at the electrode surface accompanying changes in the concentration of the analyte of interest. Choice of a technique therefore, depends on the combination of electrode, electrolyte, nanointerface and capture agent employed.
The improved performance of these methods is due to the used nanomaterials or the improved instruments or other factors?
The major contributor to the improved sensitivity is from the nanointerface materials and the enzymes/antibodies used for specific detection while the contribution by advancedinstrumentation and techniques are minimal.
The style of the references needs to be unified.
We thank the reviewer for the suggestion. The reference style has been uniformly unified in the revised manuscript.

Reviewer 4 Report
DearEditor,
this review deals with a topic of sure interest for the reference scientific community and evidently for readers of Sensors. The topic is well treated and complete. Therefore it is my opinion that the work can be accepted with minor revisions. In particular, for completeness I suggest to add the following references:
F Milano et al., Phosphate Modified Screen Printed Electrodes By Lift Treatment For Glucose Detection. Biosensors 8 (4), 91
Mr Guascito et al., Te Oxide Nanowires As Advanced Materials For Amperometric Nonenzymatic Hydrogen Peroxide Sensing. Talanta 115, 863-869
Author Response
Reviewer 4
This review deals with a topic of sure interest for the reference scientific community and evidently for readers of Sensors. The topic is well treated and complete. Therefore it is my opinion that the work can be accepted with minor revisions.
In particular, for completeness I suggest to add the following references:
Oladejo. Overview of The Metabolic Syndrome; An Emerging Pandemic Of Public Health SignificanceAnn Ib Postgrad Med. 2011 Dec; 9(2): 78–82. Pmcid: Pmc4111026. Pmid: 25161488
F Milano et al., Phosphate Modified Screen Printed Electrodes By Lift Treatment For Glucose Detection. Biosensors 8 (4), 91
Mr Guascito et al., Te Oxide Nanowires As Advanced Materials For Amperometric Nonenzymatic Hydrogen Peroxide Sensing. Talanta 115, 863-869
We thank the reviewer for the suggestion. The references suggested have been suitably included in the revised version of the manuscript.

Round 2
Reviewer 1 Report
All my concerns and suggestions were clearly attended. I suggest the paper could be published